# Topological data analysis of monopole current networks in $U(1)$ lattice gauge theory

Xavier Crean[1*], Jeffrey Giansiracusa[2†] and Biagio Lucini[1‡]

**1** Department of Mathematics, Swansea University, Bay Campus, Swansea, SA1 8EN, UK
**2** Department of Mathematical Sciences, Durham University,
Upper Mountjoy Campus, Durham, DH1 3LE, UK

* 2237451@swansea.ac.uk , † jeffrey.giansiracusa@durham.ac.uk , ‡ b.lucini@swansea.ac.uk

## Abstract

In 4-dimensional pure compact $U(1)$ lattice gauge theory, we analyse topological aspects of the dynamics of monopoles across the deconfinement phase transition. We do this using tools from Topological Data Analysis (TDA). We demonstrate that observables constructed from the zeroth and first homology groups of monopole current networks may be used to quantitatively and robustly locate the critical inverse coupling $\beta_c$ through finite-size scaling. Our method provides a mathematically robust framework for the characterisation of topological invariants related to monopole currents, putting on firmer ground earlier investigations. Moreover, our approach can be generalised to the study of Abelian monopoles in non-Abelian gauge theories.

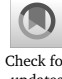

# 1 Introduction

Providing a consistent explanation of confinement in non-Abelian lattice gauge theory has been a sought-after objective for more than 50 years. Prominent avenues of research involve the analysis of two topological objects, respectively centre vortices and monopoles. Ever since Polyakov showed in [1] that, in the 3-dimensional compact QED vacuum, a monopole gas provides a consistent Abelian linear confinement mechanism by screening electric probes, there has been significant interest in the Abelian monopole picture of confinement. Mandelstam [2] and 't Hooft [3] suggested that a dual superconductor model of confinement may be formulated such that a dual Meissner effect expels electric flux lines from the bulk except for collimated tubes between charges. As suggested by Wilson in [4], in the case of 4-dimensional pure compact QED, this manifests as a bulk phase transition between a confining phase and a Coulomb phase. At strong coupling, Cooper pairs of magnetic monopoles condense and thus induce a dual Meissner effect such that electric charges are confined [5, 6]. DeGrand and Toussaint's pioneering work in [7] outlined a way to measure monopoles in lattice simulations and demonstrated the existence of a phase transition involving monopoles with critical inverse coupling $\beta_c \approx 1.01$. Subsequent work (e.g., Refs. [8, 9]) highlighted the role of monopoles in the phase transition. Further, by considering the Dirac sheets, Kerler et al. confirmed in [10] that the transition may be characterised as a percolation-type transition where, in the confining phase, monopole currents fill the volume in all four spacetime directions. An order parameter based on monopole condensation was formulated and studied in [11], while a more recent investigation of the phases of the model and of the role that monopoles play in its dynamics is provided in [12]. A different approach to the transition based on observables inspired by the XY model is provided in [13].

Following the first numerical explorations of the model [14–18], even though the dynamics of the system became well understood and the existence of a phase transition unambiguously demonstrated, the order of this phase transition was debated for a long time [19–29]. Simulations that involve small lattice sizes obscure the double-peak structure of action histograms and thus make it difficult to identify the order of the transition. Whereas, for larger lattice sizes, there exists a limitation in Monte Carlo simulations known as critical slowing down caused by severe meta-stabilities in the critical region of the transition due to the condensation of monopoles. Monopole currents self-organise in large percolating networks that wrap around a lattice with periodic boundary conditions, forming potential barriers between states of the system, and these cost a large amount of energy to break apart. In the critical region, as the lattice size increases, the probability of tunnelling between confining and Coulomb phases becomes exponentially suppressed, which means that Markov chain simulations can take a very long time to sample the configuration space sufficiently to precisely estimate $\beta_c$. Only relatively recently, with the emergence of high-performance computing and specialised numerical methods, has this been confirmed as a weak first order transition [30, 31].

When one considers percolating current networks and their unwinding mechanism, framed in this way, the problem of monopole condensation has a topological nature. On a finite-size, periodically closed lattice, monopole current lines necessarily form closed loops, which can either be global (wrapping around a direction on the hyper-torus) or local (non-wrapping), and can be connected in large, self-intersecting networks or be separated into distinct, small networks. Kerler et al. in [22] examined the winding of the monopole current networks around the spacetime torus and were able to see the signal of the phase transition. However, it was found (e.g., in [29]) that the transition is also present on a lattice representation of the 4-sphere where non-trivial winding is not possible since any loop is contractible. This left open the question of whether a topological characterisation of the phases exists independent of global winding.

More generally, topological phase structure has been studied in a variety of lattice theories. In the generalised Ising model, Blanchard et al. in [32] found a sharp transition in the Euler characteristic and Betti numbers of a cubical complex constructed by considering clusters of aligned spins. Recently, Topological Data Analysis (TDA), a field combining algebraic topology and data science, has materialised as a range of tools for analysing topological structures in datasets in a highly interpretable way. This has been used to probe various physical models and associated critical phenomena [33–46] (a useful survey is [47]) including the confinement-deconfinement phase transition in non-Abelian lattice gauge theory (and related effective models) [48–53]. Typically, topological structure in a given lattice configuration is analysed by computing the Betti numbers of a purpose-built hierarchy of cubical complexes (called a filtration) in a process known as *persistent homology*. In the case of 4-dimensional $SU(2)$ lattice gauge theory, persistent homology allowed Sale et al. in [51] to extract the critical exponents of the continuous deconfinement transition by computing the homology associated to structures built from 2-dimensional vortex surfaces.

In a similar vein, we approach the well-studied discontinuous deconfinement transition in 4-dimensional compact $U(1)$ lattice gauge theory but, since monopole currents are oriented 1-dimensional strings, we can use a simpler, readily computable application of TDA. Our novel contribution is to demonstrate that observables constructed explicitly from the homology of monopole current networks, when considered as directed graphs, allow the critical inverse coupling $\beta_c$ to be estimated robustly. Compared to the work of Kerler et al. [10], where one must compute Dirac sheets of minimal area via annealing, our methodology is a computationally more efficient way to systematically analyse the topological phase structure of a lattice configuration. We design our methodology with the broader aim of generalisation to the Abelian confinement mechanisms in non-Abelian Yang-Mills lattice gauge theory. We are motivated by the relative computational feasibility and interpretability that TDA provides. Thus, our work may be seen as a preliminary step in the application of computational topology tools to this area.

The paper is structured as follows. In Section 2, we provide a background on 4-dimensional pure compact $U(1)$ lattice gauge theory; we then introduce the DeGrand-Toussaint prescription of Dirac monopoles and the deconfinement phase transition formulated in terms of monopole currents. In Section 3, we include a background on homology and explain how topological observables constructed from monopole current networks may be used to quantitatively analyse the phase structure of a lattice configuration and present our results for estimating the critical inverse coupling $\beta_c$ of the phase transition. In Section 4, we conclude and discuss potential future directions for our work including the application to non-Abelian Yang-Mills theory. In the appendices, we provide a commentary on Dirac sheets, additional technical details and robustness checks on our scaling analysis.

## 2 $U(1)$ lattice gauge theory

### 2.1 The model

We consider pure 4-dimensional lattice gauge theory with gauge group $U(1)$. To establish notation, our lattice is $\Lambda = \{1, \ldots, L\}^4$ with periodic boundary conditions, so it is a discretisation of the 4-torus $T^4 = S^1 \times S^1 \times S^1 \times S^1$. If $\mu = 0, \ldots, 3$ indexes directions, then links are indexed by pairs $(x \in \Lambda, \mu)$ connecting sites $x$ and $x + \hat{\mu}$, and plaquettes are indexed by a lattice site $x \in \Lambda$ and a pair of directions $\mu\nu$. The gauge field $U$ assigns an element $U_\mu(x) \in U(1)$ to each link. We represent elements of $U(1)$ by angular parameters $\theta \in (-\pi, \pi]$, so $U_\mu(x) = e^{i\theta_\mu(x)}$.

The Wilson loop around the plaquette $(x, \mu\nu)$ is

$$U_{\mu\nu}(x) \equiv U_\mu(x)U_\nu(x+\hat{\mu})U_\mu^*(x+\hat{\nu})U_\nu^*(x), \tag{1}$$

where $U_\mu^*(x)$ is the complex conjugate of $U_\mu(x)$. In terms of the angular variables, we have $U_{\mu\nu}(x) = e^{i\theta_{\mu\nu}(x)}$ with $\theta_{\mu\nu}(x) = \theta_\mu(x) + \theta_\nu(x+\hat{\mu}) - \theta_\mu(x+\hat{\nu}) - \theta_\nu(x)$. Given a gauge field configuration $U$, one then defines the Wilson action in terms of the angular parameters as

$$S = \beta \sum_{x, \mu < \nu} \left(1 - \cos[\theta_{\mu\nu}(x)]\right), \tag{2}$$

where $\beta = \frac{1}{g^2}$ with $g$ the gauge coupling such that in the classical continuum limit, where the lattice spacing $a \to 0$, we have the Yang-Mills action [4].

The partition function is a path integral of the form

$$Z = \int \mathcal{D}\theta\, e^{-S(\theta)}, \tag{3}$$

with the path integral measure given by a product of Haar measures,

$$\mathcal{D}\theta = \prod_{x,\mu} d\theta_\mu(x). \tag{4}$$

The vacuum expectation value of an observable $O$ is defined as

$$\langle O \rangle = \frac{1}{Z} \int \mathcal{D}\theta\, O(\theta) e^{-S(\theta)}. \tag{5}$$

In practice, we generate $N$ samples via Monte Carlo simulation and compute the sample observables $\{O_i\}_{i=0}^{N-1}$. We may then estimate $\langle O \rangle$ by the mean of the sample observables:

$$\langle O \rangle \approx \frac{1}{N} \sum_{i=0}^{N-1} O_i. \tag{6}$$

### 2.2 Topology of monopole currents and phase structure of the model

The phase transition in $4d$ compact $U(1)$ is mediated by the condensation of monopoles, such that, in the strong coupling regime $\beta < \beta_c$, we have an electrically confining phase and, in the weak coupling regime $\beta > \beta_c$, we have a free Maxwell theory. We shall refer to the confining phase as the low-$\beta$ phase and the deconfined phase as the high-$\beta$ phase.

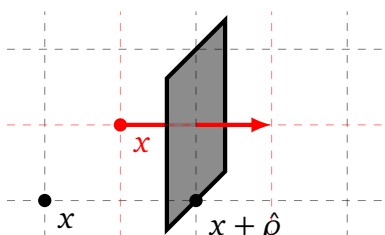

Figure 1: Given $x + \hat{\rho} \in \Lambda$, to visualise a dual lattice 4-current $j_\rho(x) = +1$ piercing a lattice 3-cube $M_\rho(x + \hat{\rho})$, we consider a three-dimensional analogue where a lattice 2-cube (grey) is pierced by a dual lattice 1-cube (red).

In a single time slice, gauge invariant monopoles are the endpoints of 1-dimensional non-gauge invariant singularities called Dirac strings (that can be thought of as infinitely thin solenoids). Given a lattice gauge field configuration, we can measure Dirac strings by noting that they carry units of $2\pi$ flux. Since plaquette variables represent the electromagnetic field strength tensor, i.e., $\theta_{\mu\nu}(x) = a^2 F_{\mu\nu}(x) + O(a^3)$ with lattice spacing $a$, they measure the magnetic flux of the corresponding $1 \times 1$ square area of the plaquette. Since link variables $\theta_\mu(x)$ take values in the interval $(-\pi, \pi]$, the plaquette variables $\theta_{\mu\nu}(x)$ are valued in $(-4\pi, 4\pi]$; the physical plaquette angle is defined as

$$\bar{\theta}_{\mu\nu}(x) \equiv \theta_{\mu\nu}(x) - 2\pi n_{\mu\nu}(x) \in [-\pi, \pi], \tag{7}$$

where $n_{\mu\nu}(x) \in \{0, \pm1, \pm2\}$ is interpreted as the number of Dirac strings through the plaquette [7]. The gauge invariant monopole number in any given 3-cube may be computed by counting the net number of Dirac strings entering/exiting the volume. Thus, for each lattice site $x \in \Lambda$, we have four monopole numbers given by the relation

$$M_\rho(x) = \frac{1}{4\pi}\varepsilon_{\rho\sigma\mu\nu}[\bar{\theta}_{\mu\nu}(x + \hat{\sigma}) - \bar{\theta}_{\mu\nu}(x)] = -\frac{1}{2}\varepsilon_{\rho\sigma\mu\nu}[n_{\mu\nu}(x + \hat{\sigma}) - n_{\mu\nu}(x)] \tag{8}$$

(c.f. 3$d$ continuum Maxwell equation $\partial_i B_i(\vec{x}) = 2\pi M(\vec{x})$).

We may naturally define the dual lattice $\Lambda^*$ where 4-cubes are dual to vertices, 3-cubes dual to links, 2-cubes dual to 2-cubes etc. On $\Lambda^*$, given that links are dual to 3-cubes, monopole worldlines are represented by 4-currents $j_\rho(x) = M_\rho(x + \hat{\rho})$ that are defined on links (Figure 1). Given an appropriate orientation, we have that

1. $j_\rho(x) = 0 \implies \nexists$ current line on link at $x$,

2. $j_\rho(x) = \pm1 \implies \exists$ one current line on link at $x$,

3. $j_\rho(x) = \pm2 \implies \exists$ two current lines on link at $x$,

which must obey the conservation of current

$$\sum_\rho [j_\rho(x) - j_\rho(x - \hat{\rho})] = 0 \tag{9}$$

(c.f. $\partial_\rho j_\rho = 0$). This implies that the current lines must form a union of closed loops. Note that, in any given configuration, the lattice equivalent of Gauss' law for magnetism must hold globally. Therefore, since we have periodic and untwisted boundary conditions, the net charge of periodically closed current loops must be zero, i.e., for a given loop that wraps around a given direction on $T^4$, there must also be an oppositely oriented partner loop.

Defining a network as a connected set of current lines and a Dirac sheet as the worldsheet of a Dirac string such that the boundary of a Dirac sheet is a current loop (see Appendix A), Kerler et al. in [10] characterised the phase structure of the system as follows:

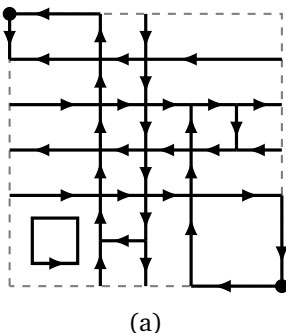

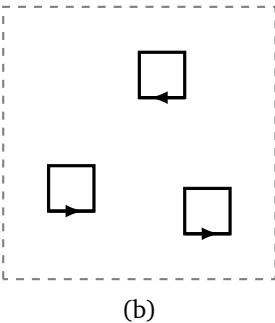

(a)                                                    (b)

Figure 2: In this schematic, we have a two-dimensional analogue of an example configuration where the dashed grey lines represent the periodic boundary of $T^2 = S^1 \times S^1$ and the black arrows represent monopole current networks. In Figure 2a, the configuration is in the low-$\beta$ phase and so has a percolating current network as well as isolated, small current networks. In Figure 2b, the configuration is in the high-$\beta$ phase and so we have only small current networks.

1. Configurations sampled from the low-$\beta$ phase ($\beta < \beta_c$) contain (with high probability) a Dirac sheet with boundary of non-trivial winding number around the torus in each of the four directions, and hence a percolating current network in each direction.

2. For configurations sampled from the high-$\beta$ phase ($\beta > \beta_c$), all Dirac sheets have a boundary that is homotopically trivial (with high probability), and hence there does not exist a percolating current network. Note that, subject to conditions, non-trivial Dirac sheets without boundary may form, giving rise to meta-stable states [19].

In the low-$\beta$ phase, whilst there is a large, entangled, percolating current network, there may also exist small networks that do not form part of the largest network, since production of a monopole-antimonopole pair is not strongly suppressed. In the high-$\beta$ phase, we only have non-percolating current loops. In Figure 2, we have a two-dimensional schematic of a configuration in the low-$\beta$/high-$\beta$ phase.

## 2.3 Estimating the critical inverse coupling $\beta_c$

There does not exist a way to compute the critical inverse coupling $\beta_c$ of the phase transition analytically. However, on finite-size lattices, one may compute the psuedo-critical coupling $\beta_c(L)$ via several different numerical methods. In the infinite volume limit $L \rightarrow \infty$, these psuedo-critical values converge to $\beta_c \approx 1.01$.

The phase transition in compact $U(1)$ has severe meta-stabilities at criticality which make generating configurations using Monte Carlo simulation a computationally expensive exercise. Simulations may become stuck in local minima of the action which take a very long time to tunnel out of. Specialised numerical methods have been developed to overcome these issues and have been able to estimate $\beta_c$ to a high degree of accuracy [54, 55].

The standard observable one uses to estimate $\beta_c$ is the average plaquette action

$$E = \frac{1}{6V} \sum_{x, \mu < \nu} \cos[\theta_{\mu\nu}(x)], \tag{10}$$

where volume $V = L^4$. Typically, one computes cumulants of $E$, such as the specific heat at constant volume

$$C_V(\beta) = 6\beta^2 V \left( \langle E^2 \rangle - \langle E \rangle^2 \right), \tag{11}$$

Table 1: Reference $\beta_c(L)$ values, i.e., locations of the specific heat $C_V(\beta)$ maxima as a function of lattice size [54, 55].

| $L$ | $\beta_c(L)$ |
| --- | --- |
| 6 | 1.001794(64) |
| 8 | 1.00744(2) |
| 10 | 1.00939(2) |
| 12 | 1.010245(1) |

at many values of $\beta$, and then locates the pseudo-critical value $\beta_c(L)$ at which $C_V$ is maximal. The estimation of the location of the peak uses histogram reweighting, which is a technique for estimating the value of an observable at many interpolating $\beta$ values; for a review of histogram reweighting, see Appendix B.1.

Using these psuedo-critical $\beta_c(L)$, we may estimate the infinite volume limit of the critical coupling $\beta_c$ via a finite-size scaling analysis given the asymptotic relation

$$\beta_c(L) = \beta_c + \sum_{k=1}^{\infty} B_k L^{-4k}, \tag{12}$$

where the sum may be truncated by parameter $k_{max} < \infty$. Further, we estimate the standard error via the bootstrapping method; for a review, see Appendix B.2.

Since this phase transition is weak first order (WFO), finite-size effects will obscure the double peak structure of action histograms for smaller lattice sizes. This is because, for a WFO transition, the critical correlation length $\xi_c$ is typically very large and, for small lattice volumes $L^4$, with $L$ the measure of one of the four equal edges of the lattice,[1] we may have $L \leq \xi_c$. In this region, relative to the lattice, $\xi_c$ is effectively infinite and the transition behaves as if it were second order. To mitigate these pretransitional effects, we only take lattice sizes $L \geq 6$, which have been shown in the literature to present a double peak structure.

The goal of our analysis is to construct robust observables using topological data analysis of monopoles to estimate $\beta_c(L)$ for a suitable choice of finite lattice sizes, i.e., $L \in \{6, 7, 8, 9, 10, 11, 12\}$, in order to perform a finite-size scaling analysis to estimate $\beta_c$ in the infinite volume limit $L \to \infty$.

We generate $N = 200$ lattice configurations using a Monte Carlo (MC) importance sampling method: each update consists of one heatbath update, consisting of an approximate heatbath step corrected by a Metropolis step [56], and five over-relax updates. Computing estimates of $\beta_c$, with low-statistics using this algorithm, of comparable accuracy and precision to literature reference values (see Table 1) would be very computationally expensive and is not our objective. Instead, we take a computationally feasible number of MC updates for each respective lattice size, i.e., $\geq 8.4 \times 10^6$, and compute the $\beta_c$ given by the average plaquette observable $E$ which we expect to be close to the reference values but with a small systematic bias that can be quantified by comparing with the literature. We will then use the observable $E$ as a benchmark for our new topological observables, i.e., if a topological observable returns an estimate of $\beta_c$ statistically consistent with the fiducial value estimated via $E$, we claim that the topological observable acts as a phase indicator for the phase transition.

---

[1] In this study, we will only consider 4-cubical lattices, indicating their linear size with $L$.

# 3 Topological data analysis

As described in Section 2.2, a configuration in the confining phase contains a percolating monopole current network, whilst a configuration in the Coulomb phase does not. We construct simple topological observables from monopole current networks to estimate the critical inverse coupling $\beta_c$ via a finite-size scaling analysis.

## 3.1 Background on homology

We give a brief review of the main ideas behind homology; then, we show how to construct the zeroth and first homology groups of a directed graph. For a more complete review, see [57].

### 3.1.1 Chain complex

In algebraic topology, *homology* is a way to make precise the notion of a $k$-dimensional "hole" by way of an algebraic structure called a *chain complex*: given a sequence of vector spaces $\{C_k\}_{k=0}^n$ each equipped with respective *boundary operator* $\partial_k : C_k \to C_{k-1}$ such that $\partial_k \circ \partial_{k+1} \equiv 0$, then we have the chain complex

$$\cdots \xrightarrow{\partial_{k+1}} C_k \xrightarrow{\partial_k} C_{k-1} \xrightarrow{\partial_{k-1}} \cdots. \tag{13}$$

The *image* and *kernel* of $\partial_k$ are defined as

$$\operatorname{im}\partial_k \equiv \{c' \in C_{k-1} \,|\, \partial_k(c) = c', c \in C_k\}, \tag{14}$$

$$\ker\partial_k \equiv \{c \in C_k \,|\, \partial_k(c) = 0\}, \tag{15}$$

and we have that

$$\operatorname{im}\partial_{k+1} \subseteq \ker\partial_k. \tag{16}$$

We refer to elements in $\operatorname{im}\partial_{k+1}$ as *boundaries* and elements in $\ker\partial_k$ as *cycles*. The *$k$-th homology group* is the quotient group

$$H_k \equiv \ker\partial_k / \operatorname{im}\partial_{k+1}. \tag{17}$$

It measures the difference between these subspaces, i.e., the failure of Equation (16) to be an equality. Non-zero elements correspond to cycles that are not boundaries.

In practice, the chain complex is constructed from a geometric object such as a simplicial or cubical complex; $C_k$ is the vector space with basis given by the $k$-dimensional cells. In this context, the dimension of $H_k$ is known as the *$k$-th Betti number $b_k$*, and it counts the number of $k$-dimensional holes.

### 3.1.2 Homology of a graph

Graphs have a natural interpretation as chain complexes. This enables us to apply the concepts discussed in the previous section to graphs. A directed graph $X$ is defined as the tuple $(V, E)$ comprising a vertex set $V$ and an edge set $E = \{(u, v) \,|\, u, v \in V\}$ consisting of ordered pairs of vertices (c.f. a simplicial complex). From a graph, we produce a short chain complex

$$0 \to C_1 \xrightarrow{\partial_1} C_0 \to 0, \tag{18}$$

by defining $C_0$ to be the vector space with basis given by the vertices, and similarly $C_1$ has basis given by the edges. The boundary map sends an edge $(u, v)$ to $u - v$ and is extended to all of $C_1$ by linearity. Note that maps into and out of the zero vector space are necessarily the

zero map and thus implicit in Equation (18). We may subsequently compute the zeroth and first homology groups of the graph $X$,

$$
\begin{aligned}
H_0(X) &= C_0/\operatorname{im}\partial_1\,, \\
H_1(X) &= \ker\partial_1\,.
\end{aligned}
\tag{19}
$$

For graphs, the ranks of the homology groups are independent of the choice of coefficient field.[2] In the following methodology, our chosen coefficient field is $\mathbb{Z}_2$ for computational simplicity.

The Betti number $b_0(X) \equiv \dim H_0(X)$ is the number of connected components of the graph and the Betti number $b_1(X) \equiv \dim H_1(X)$ is the number of loops. For details on how we explicitly compute the Betti numbers of graph $X$, see Appendix B.3.

## 3.2 Analysis of monopole current networks

### 3.2.1 Betti number observables

In a given lattice gauge field configuration, we may naturally endow the disjoint union of monopole current networks with the structure of a (possibly disconnected) directed graph. This is achieved by associating dual lattice links with $j_\rho(x) = \pm 1, \pm 2$ to directed graph edges. We must assign a consistent choice of orientation to each edge in our directed graph so that our definition of graph homology is concordant with that of Section 3.1.2. However, the resulting homology of the graph is independent of this choice. After assigning an orientation to each edge, we then include its boundary vertices to complete the graph. Thus, we have a directed edge set $E$ constructed from current lines and a vertex set $V$ consisting of their boundary vertices. We denote this monopole current graph by $X_j = (V, E)$.

By Equation (9), current networks are necessarily unions of closed loops. Note that these current loops may intersect. We compute the Betti numbers $b_0(X_j) = \dim H_0(X_j)$, the number of connected components of $X_j$, and $b_1(X_j) = \dim H_1(X_j)$, the number of loops of $X_j$ (for details, see Appendix B.3). As mentioned previously, we use the coefficient field $\mathbb{Z}_2$ but, since this is a 1-dimensional complex, the resulting Betti numbers are independent of the coefficient field.

To construct more useful observables from $b_0$ and $b_1$ that can be directly compared across various lattice sizes, we normalise by the number of lattice sites, i.e., we divide by the volume $V = L^4$. Thus, we compute the mean density of distinct networks $\rho_0 \equiv \langle b_0 \rangle / V$ and the mean density of loops $\rho_1 \equiv \langle b_1 \rangle / V$. Note that, given $N = 200$ samples, as per Equation (6), we estimate $\langle b_k \rangle$ by the sample mean. For both $k = 0, 1$, we compute the normalised variance of the Betti numbers

$$
\chi_k \equiv (\langle b_k^2 \rangle - \langle b_k \rangle^2)/V\,.
\tag{20}
$$

We shall use $\chi_k$ to estimate the pseudo-critical inverse coupling $\beta_c(L)$ by locating the maximum of the reweighted variance curve – reweighting allows for a more precise estimate of the location of the maximum.

---

[2]This follows from the Universal Coefficient Theorem and the fact that a graph is homotopy equivalent to a finite wedge sum of circles: Consider a spanning tree of any given graph $X$; this can be contracted to a point such that we are left with a bouquet of circles. The homology groups of a bouquet of $n$ circles $\bigvee_{i=1}^{n} S^1$ can be easily computed: $H_0(\bigvee_{i=1}^{n} S^1; \mathbb{Z}) = \mathbb{Z}$, $H_1(\bigvee_{i=1}^{n} S^1; \mathbb{Z}) = \mathbb{Z}^n$ and $H_k(\bigvee_{i=1}^{n} S^1; \mathbb{Z}) = 0$ for $k \geq 2$. These are free Abelian groups and thus torsion free. By the Universal Coefficient Theorem (e.g., see page 129 in Ref. [57]), we have that $H_k(X; \mathbb{Z}) \otimes A \simeq H_k(X; A)$ where $A$ is any Abelian group. Thus, $H_0(X; A) \simeq A$, $H_1(X; A) \simeq A^n$ and $H_k(X; A) = 0$ for $k \geq 2$ so the rank of the homology group is independent of $A$. One may extend this analysis to the case where coefficients are taken as field $\mathbb{F}$ and $n$ represents the dimension of the homology vector space.

### 3.2.2 Numerical results and interpretation

As outlined in Refs. [10,22], in the high-$\beta$ phase ($\beta > \beta_c$), the currents typically form a disjoint collection of small loops and so we expect $b_1 \approx b_0$. In the low-$\beta$ phase, one typically sees a large percolating network consisting of many intersecting loops together with a few isolated small networks and so we expect $b_1 > b_0$.

As seen in Figure 3, the density $\rho_0$ reveals the signature of the phase transition; this is compatible with the accepted physical picture seen in the literature (e.g., Ref. [22]). In the low-$\beta$ phase, the probability that small networks form alongside the large percolating network is non-zero; therefore, we have a small but non-zero number of connected components. As $\beta$ approaches the critical value from below, $\rho_0$ reaches a maximum value as the large percolating network is broken into smaller networks. As $\beta$ further increases into the high-$\beta$ phase, the probability of sampling a configuration with a large number of monopole current loops decreases, hence the expected number of connected components decreases. A scatter plot of $\chi_0$ is provided in Figure 4. For all lattice sizes $L = 6, ..., 12$, we find that the reweighted $\chi_0$ curves peak at the critical point allowing us to extract the pseudo-critical $\beta_c(L)$ values.

As seen in Figure 5, we find that the observable $\rho_1$ acts as a phase indicator since the number of loops is much greater in the low-$\beta$ phase than in the high-$\beta$ phase, with a transition about the critical value. Again, this is compatible with the physical picture seen in the literature. A scatter plot of $\chi_1$ is provided in Figure 6. In the low-$\beta$ phase, we find a relatively large variance compared to the high-$\beta$ phase. We hypothesise this is due to the fact that, when in the low-$\beta$ phase, we sample from a Boltzmann distribution that has a large variation of configurations with percolating networks from which one can sample, each with a different number of self-intersections. Whereas, in the high-$\beta$ phase, one samples configurations with small disconnected networks that are unlikely to self-intersect leading to a lower variation in the measurement of $b_1$. On the smaller lattices with $L \leq 9$, the peak of the variance curve is partially obscured meaning that the location of the peak of the reweighted variance curves of $\rho_1$ is less evident.

We present our estimated pseudo-critical $\beta_c(L)$ values for $E$, $\rho_0$ and $\rho_1$ in Table 2.[3] Reweighting windows have been suitably selected around the peaks in the variance plots and error bars are computed using the bootstrap method with $N_{bs} = 500$. Further, we perform a finite-size scaling analysis, using the parametrisation from Equation (12), to estimate $\beta_c$ in the infinite volume limit via a polynomial regression setting $k_{\max}$ accordingly; see Appendix C for suitable robustness checks. Our estimates for the infinite volume critical inverse coupling $\beta_c$ are in Table 3 and demonstrate statistical consistency.

As seen in Figure 7, we find that in the low-$\beta$ phase $\rho_1$ is negatively correlated with $\rho_0$ and in the high-$\beta$ phase $\rho_1$ is positively correlated with $\rho_0$; there exists a turning point located in the critical region. This fits with the general physical picture seen in the literature (e.g., Ref. [22]). In the low-$\beta$ phase, we have a large self-intersecting, percolating network and a non-zero number of small independent networks. As $\beta$ decreases, the size of the largest network increases; therefore, the number of self-intersections increases $\uparrow b_1$ and the number of distinct networks decreases $\downarrow b_0$. In the critical region, we have a maximum number of distinct networks as the large percolating network breaks into smaller networks. In the high-$\beta$ phase, we have small networks which are unlikely to self-intersect, thus $b_1$ evolves proportionally to $b_0$.

---

[3]It is important to note that, whilst pseudo-critical $\beta_c(L)$ values for the observables $E$, $\rho_0$, $\rho_1$ are not equal for finite size $L$, which is to be expected, they converge in the physically significant $L \to \infty$ limit (see Table 3).

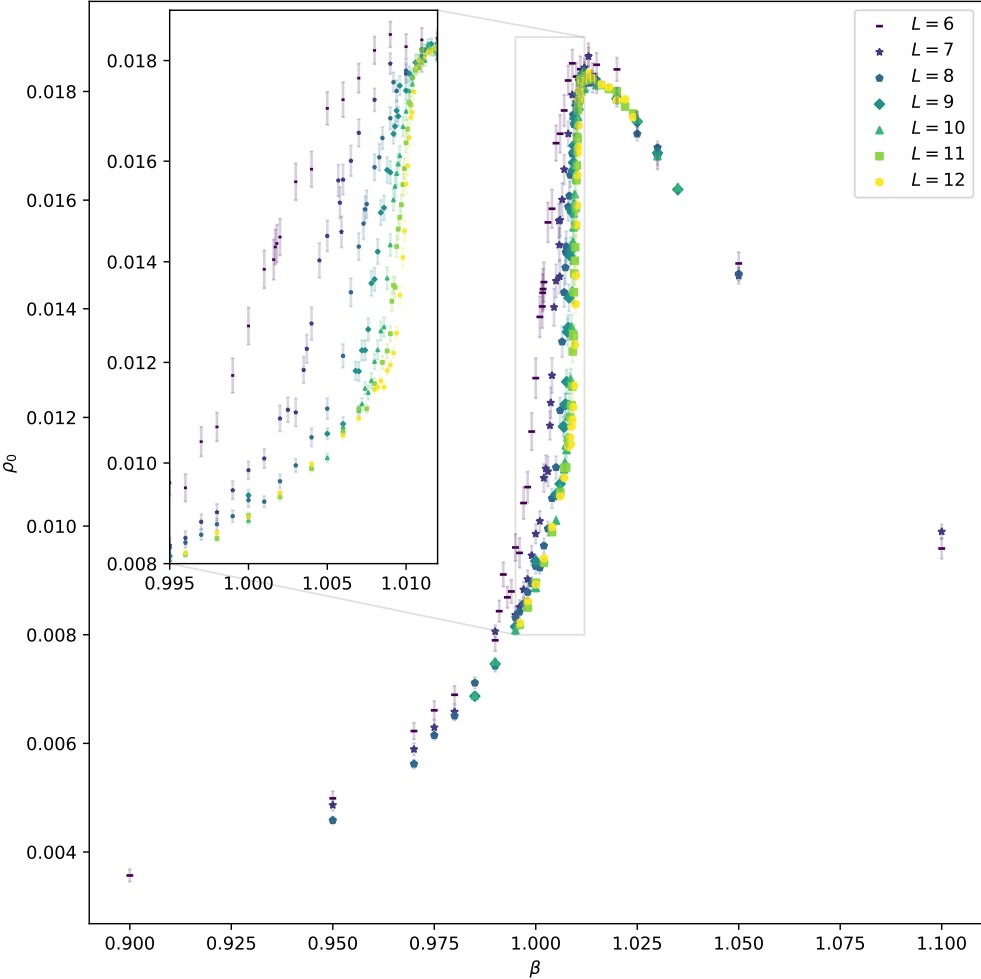

Figure 3: **(main plot)** A scatter plot with error bars of the mean density of networks observable $\rho_0$ against $\beta$ for each respective lattice size $L \in \{6, ..., 12\}$. Error bars are computed via bootstrapping with $N_{bs} = 500$. The mean density of networks $\rho_0$ captures the signature of the transition and corresponds with the picture seen in the literature: a low-$\beta$ phase with few connected components correspondingly $\rho_0$ is small, a critical region where the largest network is broken into smaller components correspondingly $\rho_0$ obtains its maximum, and a high-$\beta$ phase with a number of small networks that decreases as $\beta$ increases correspondingly $\rho_0$ decreases. **(inset plot)** A zoom-in of the same plot around the critical region showing a sharper transition as the lattice size $L$ increases.

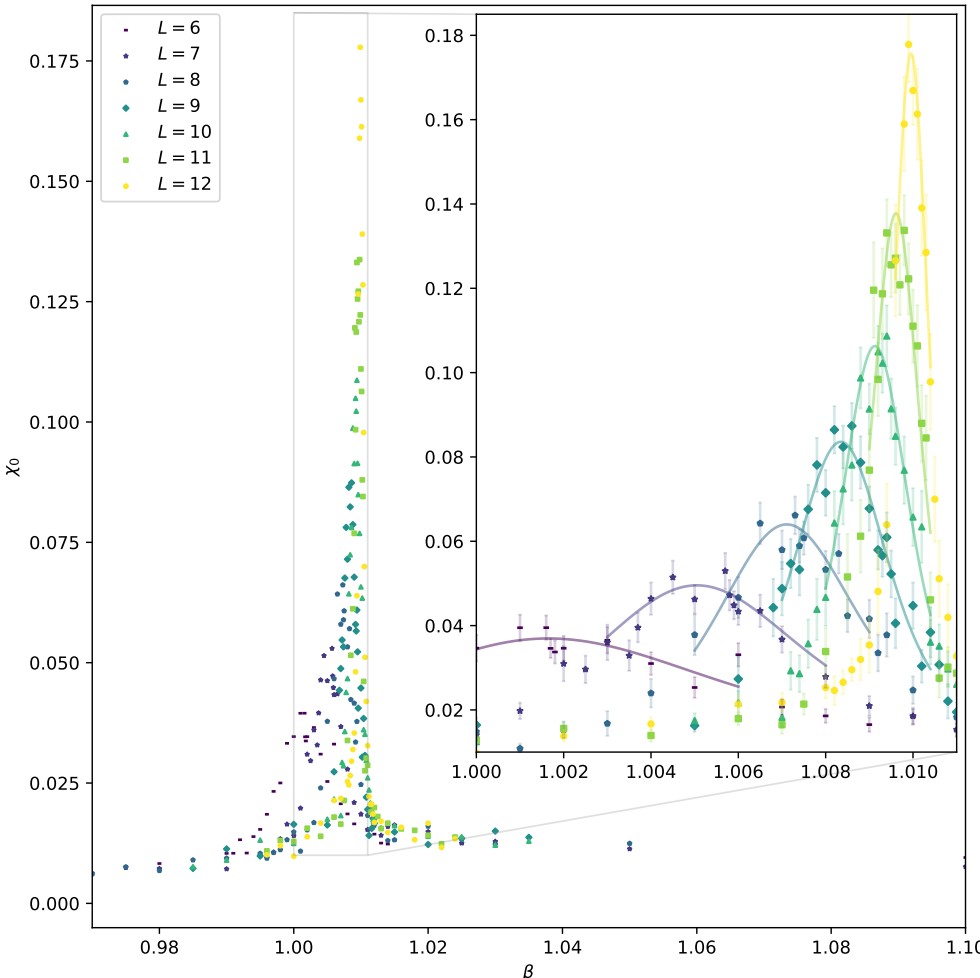

Figure 4: **(main plot)** A scatter plot of the normalised variance $\chi_0$ against $\beta$ for each respective lattice size $L \in \{6, ..., 12\}$. Error bars have been omitted from the main plot for ease of viewing. One can see that $\chi_0$ peaks in the critical region indicative of the tunnelling between phases expected for a first order phase transition. **(inset plot)** A zoom-in of the same plot around the critical region with error bars and reweighting curves included. Error bars are computed via bootstrapping with $N_{bs} = 500$. Reweighting windows have been suitably selected around each respective peak which, as the lattice size $L$ increases, can be seen to be taller and more tightly concentrated around the respective pseudo-critical $\beta_c(L)$. The reweighting procedure allows for a more precise estimate of $\beta_c(L)$ to be produced.

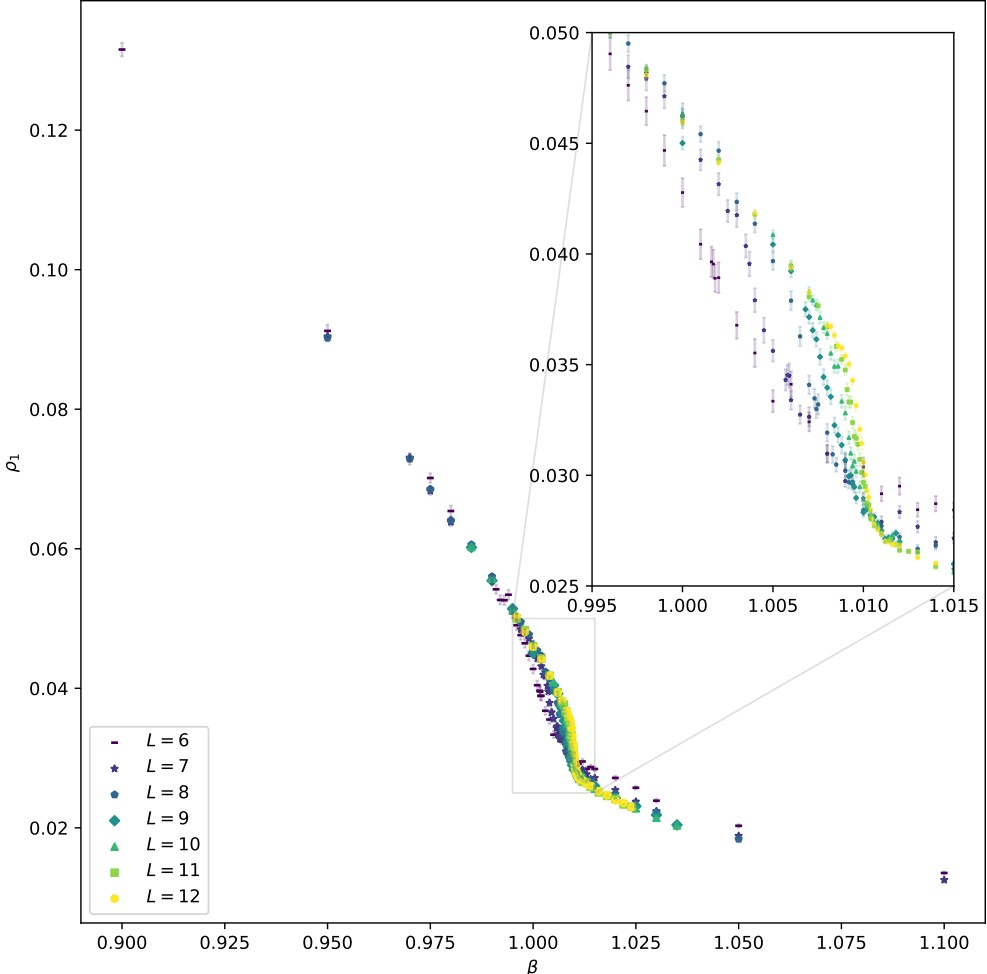

Figure 5: **(main plot)** A scatter plot with error bars of the mean density of loops observable $\rho_1$ against $\beta$ for each respective lattice size $L \in \{6, ..., 12\}$. Error bars are computed via bootstrapping with $N_{bs} = 500$. One can see that the mean density of loops $\rho_1$ decreases dramatically in the critical region which, again, is concordant with the physical picture seen in the literature. **(inset plot)** A zoom-in of the same plot around the critical region showing a sharper transition as the lattice size $L$ increases.

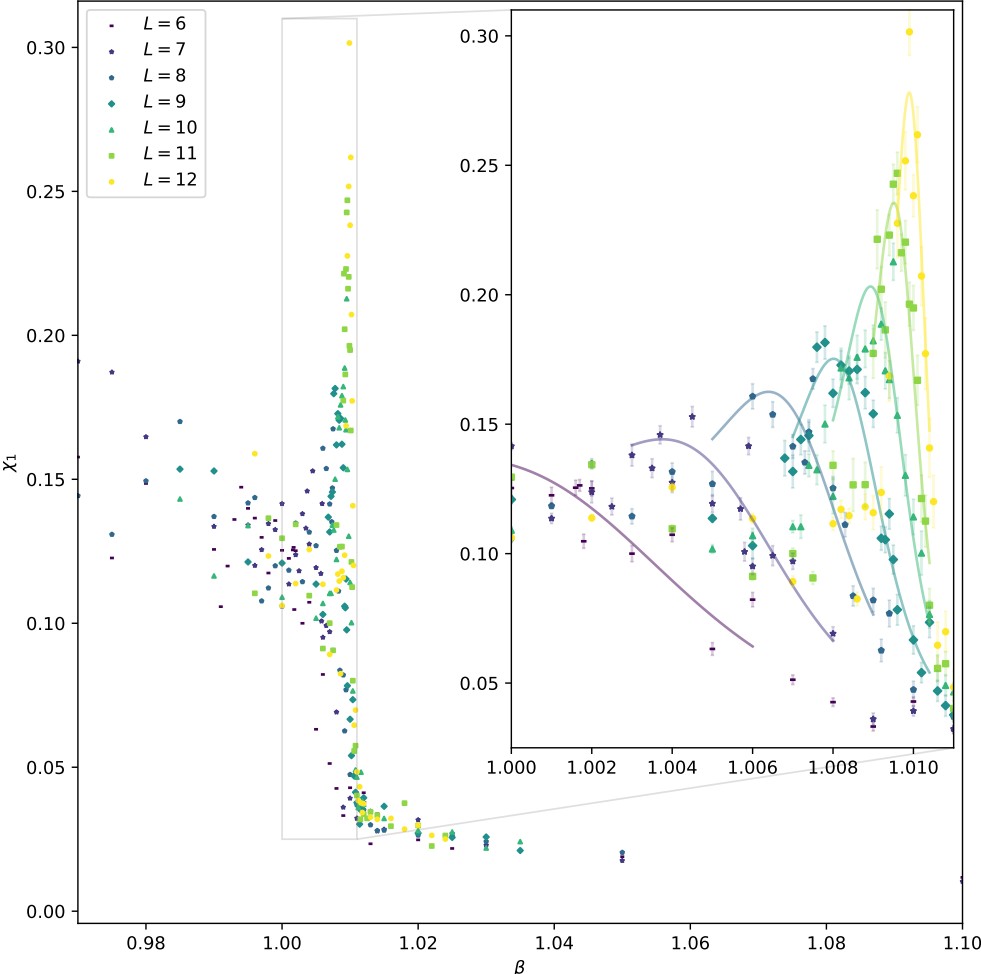

Figure 6: **(main plot)** A scatter plot of the normalised variance $\chi_1$ against $\beta$ for each respective lattice size $L \in \{6, ..., 12\}$. Error bars have been omitted from the main plot for ease of viewing. One can see that $\chi_1$ peaks in the critical region indicative of the tunnelling between phases expected for a first order phase transition. However, in the low-$\beta$ phase, there exists a relatively large variance especially for the smaller lattice sizes. For lattice sizes $L \leq 9$, the peak of $\chi_1$ is partially obscured by the large variance values in the low-$\beta$ phase. For lattice sizes $L \geq 10$, the peak of the variance is clear enough to easily locate, even with a relatively large variance in the low-$\beta$ phase. **(inset plot)** A zoom-in of the same plot around the critical region with error bars and reweighting curves included. Error bars are computed via bootstrapping with $N_{bs} = 500$. Reweighting windows have been suitably selected around each respective peak which, as the lattice size $L$ increases, can be seen to be taller and more tightly concentrated around the respective pseudo-critical $\beta_c(L)$. The reweighting procedure allows for a more precise estimate of $\beta_c(L)$ to be produced.

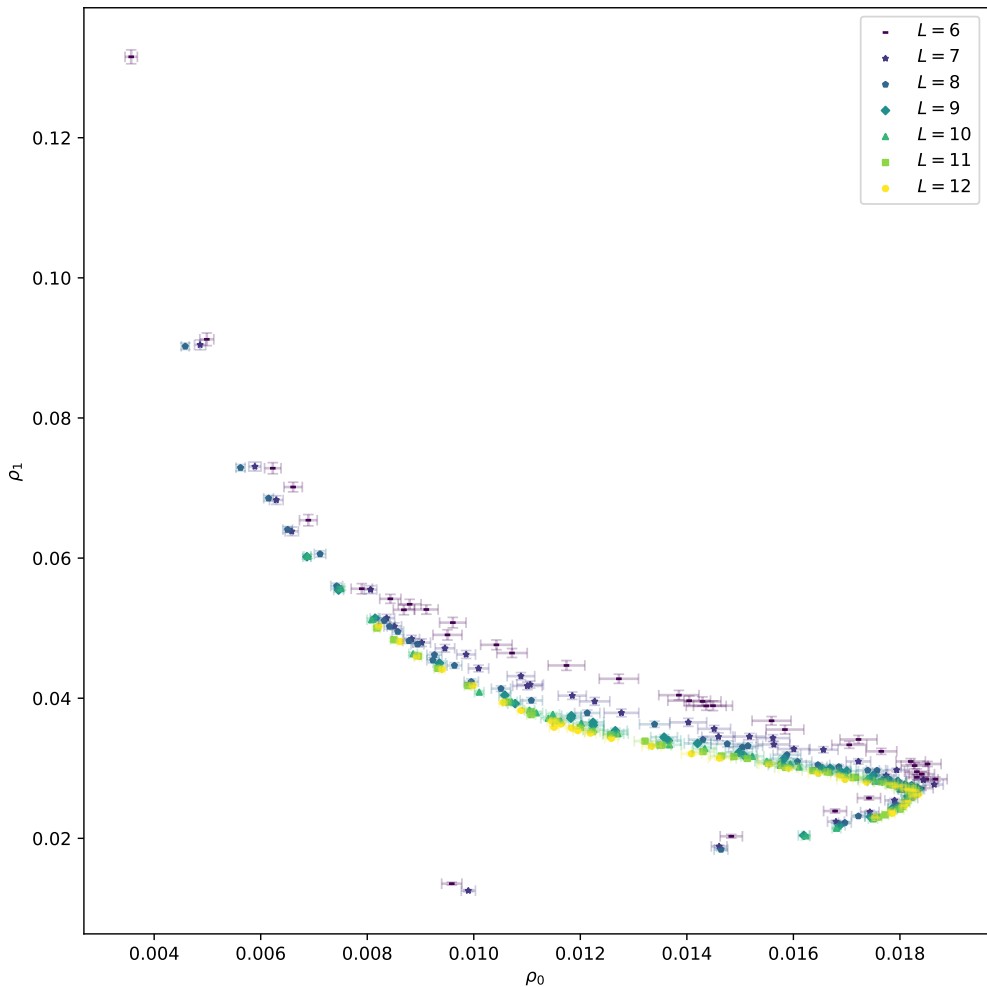

Figure 7: A scatter plot of $\rho_1$ against $\rho_0$ for the range $\beta \in [0.9, 1.1]$ such that the top left of the plot corresponds to the low-$\beta$ phase, the centre of the plot corresponds to the critical region and the bottom the high-$\beta$ phase. In the high-$\beta$ phase, one can see a strong positive correlation between $\rho_0$ and $\rho_1$ as one would expect for distinct, small networks.

Table 2: The first column gives the lattice size $L$. The second column gives the pseudo-critical $\beta_c(L)$ computed by locating the peak of the reweighted variance curve of the plaquette operator $E$; as we expect our configurations to be systematically biased, we present this as the fiducial value. The third and fourth columns give the pseudo-critical $\beta_c(L)$ computed by locating the peak of the reweighted variance curve of $\rho_0$ and $\rho_1$ respectively.

| | $\beta_c(L)$ | | |
|---|---|---|---|
| $L$ | $E$ | $\rho_0$ | $\rho_1$ |
| 6 | 1.0015(1) | 1.0016(1) | 0.9985(6) |
| 7 | 1.00502(6) | 1.0051(2) | 1.0038(1) |
| 8 | 1.00706(5) | 1.00710(9) | 1.00641(7) |
| 9 | 1.00831(3) | 1.00834(3) | 1.00803(4) |
| 10 | 1.00912(2) | 1.00914(2) | 1.00892(2) |
| 11 | 1.00961(2) | 1.00961(2) | 1.00951(2) |
| 12 | 1.00995(1) | 1.00996(1) | 1.00989(1) |

Table 3: For $E$, $\rho_0$ and $\rho_1$, the estimated infinite volume critical inverse coupling $\beta_c$ with error bars computed via bootstrapping.

| $E$ | $\rho_0$ | $\rho_1$ |
|---|---|---|
| 1.01071(3) | 1.01076(6) | 1.01076(6) |

## 4 Conclusion and discussion

In this paper, we have constructed two novel observables that may be used to analyse the phase structure of compact $U(1)$ lattice configurations. These are built from topological invariants of monopole current networks and we demonstrate their viability as pseudo-order parameters by showing that they may be used to extract the critical inverse coupling $\beta_c$ such that results yield good statistical agreement with the average plaquette observable $E$. Whilst our results are an order of magnitude less precise than the literature reference $\beta_c$, noting our low statistics $N = 200$, we do not design our methodology necessarily to achieve high precision. An advantage of our methodology is that one may probe the structure of a configuration's monopole networks, therefore, providing more evidence for a percolation-type transition of monopole currents and deeper physical insight. We claim that this provides supplementary results to [10, 22].

In order to further probe the phase structure of configurations, one might expand our analysis by using tools from TDA such as persistent homology to design more sophisticated observables, that retain a high level of interpretability, based on topological invariants of monopole current networks. This work is currently in progress and will be presented elsewhere.

As mentioned previously, going forward, we intend to expand our methodology to extract phase information from monopole currents germane to Abelian confinement mechanisms in non-Abelian Yang-Mills lattice gauge theories (as first described in Ref. [58]). More specifically, in theories where the gauge group is the special unitary group $SU(N)$ with maximal Abelian subgroup $U(1)^{N-1}$, one may perform a maximal Abelian gauge projection, as in Refs. [59–64], such that, up to gauge-fixing ambiguities (e.g., see Refs. [65–69]), the full theory may be decomposed into $N - 1$ distinct $U(1)$ theories. Each $U(1)$ theory comes with a corresponding

monopole "species". For example, in pure $SU(3)$ lattice gauge theory, which describes the strong interaction, there exist two distinct Abelian monopole species, which have been analysed previously in the literature, e.g., see Refs. [70, 71]. Note that, in our study, we have considered monopoles at temperature $T = 0$ (specified by the four equal lattice dimensions such that the time dimension size is $N_t = L$ and spatial dimension size is $N_s = L$). In these $SU(N)$ theories, it will be interesting to examine whether it is possible to quantitatively analyse, across the deconfinement phase boundary, topological structures constructed from Abelian monopole species in the $T > 0$ "thermal" regime where $N_s > N_t$.

# Acknowledgments

The authors would like to thank Tom Pritchard for advice on optimising computational resources and Ed Bennett for feedback on the figures and the accompanying software release [72]. BL wishes to thank Claudio Bonati for discussions on the Monte Carlo update algorithm used in this work. Monte Carlo simulations were performed using software developed by Claudio Bonati with contributions from Nico Battelli, Marco Cardinali, Silvia Morlacchi and Mario Papace. Topological Data Analysis was computed using giotto-tda [73]. Histogram reweighting was computed using pymbar [74].

**Author contributions**   XC devised the specific methodology and performed the numerical work. BL and JG contributed equally to the formulation of the problem and devised the general methodology. All authors contributed equally to the interpretation of the results.

**Funding information**   XC was supported by the Additional Funding Programme for Mathematical Sciences, delivered by EPSRC (EP/V521917/1) and the Heilbronn Institute for Mathematical Research. JG was supported by EPSRC grant EP/R018472/1 through the Oxford-Liverpool-Durham Centre for Topological Data Analysis. The work of BL was partly supported by the EPSRC ExCALIBUR ExaTEPP project EP/X017168/1 and by the STFC Consolidated Grants No. ST/T000813/1 and ST/X000648/1. Numerical simulations have been performed on the Swansea SUNBIRD cluster, part of the Supercomputing Wales project. Supercomputing Wales is part funded by the European Regional Development Fund (ERDF) via Welsh Government.

**Research data access statement**   The data shown in this manuscript can be downloaded from Ref. [75]. The code used to generate the Monte Carlo configurations and the analysis workflow used to produce the tables and the figures reported in this work can be downloaded from Ref. [72].

# A   Dirac sheets

If Dirac strings pass through plaquette $\theta_{\mu\nu}(x)$, the dual plaquette $\theta^*_{\rho\sigma}(x) = \frac{1}{2}\varepsilon_{\rho\sigma\mu\nu}\theta_{\mu\nu}(x)$ retains the same information through

$$p_{\rho\sigma}(x) = -\frac{1}{2}\varepsilon_{\rho\sigma\mu\nu}n_{\mu\nu}(x + \hat{\rho} + \hat{\sigma}) \in \{0, \pm 1, \pm 2\}. \tag{A.1}$$

Thus, given Equation (8), we have the dual relation

$$j_\rho(x) = \sum_\sigma [p_{\rho\sigma}(x) - p_{\rho\sigma}(x - \hat{\sigma})]. \tag{A.2}$$

One may define a Dirac sheet $S$ as the worldsheet of a Dirac string, i.e., as a 2-surface consisting of connected, oriented Dirac plaquettes such that $j_\rho(x) \neq 0$ only on the boundary $\partial S$ of the sheet, i.e., $\partial S$ is an oriented current loop. Since Dirac sheets retain information about Dirac strings, they are clearly not gauge invariant but may be classified into equivalence classes $[S]$ up to gauge transformation; $[S]$ may consequently be classified into distinct homotopy classes. If we represent $[S]$ by the sheet in the class with mininal area $S_{min}$, we may compare the minimal areas of different homotopy classes; typically, the class with the largest minimal area that spans $T^4$ is referred to as non-trivial.

It is worth reiterating once again that Dirac sheets are not gauge invariant. Thus, in a given configuration, it is very unlikely that the numerically determined Dirac sheets have minimal area – a necessity when classifying their gauge-equivalence classes by homotopy type for use as an order parameter. It is possible to find an equivalence class' representative minimal area sheet $S_{min}$ via an annealing process [10]. However, this is computationally expensive and so we shall instead focus on their gauge invariant boundaries, i.e., monopole current loops.

## B    Methods

### B.1    Histogram reweighting

In order to achieve a more precise estimation of the critical inverse coupling $\beta_c$, we make use of a standard technique in lattice field theory, to produce interpolating variance curves, called *histogram reweighting*. Given the action of a configuration $S$, we may express the ensemble average of an observable $O$ at inverse coupling $\beta$ in terms of the ensemble average at another inverse coupling $\beta'$ via

$$\langle O \rangle_\beta = \frac{\langle O e^{-(\beta - \beta')S} \rangle_{\beta'}}{\langle e^{-(\beta - \beta')S} \rangle_{\beta'}}. \tag{B.1}$$

Regarding computational tractibility, one can only successfully reweight in regions where the distributions of the sampled actions at $\beta$ and $\beta'$ overlap sufficiently. Multiple histogram reweighting allows us to exploit the fact that we have many sampled configurations at a range of inverse couplings. Given $N_i \in \mathbb{N}$ configurations each at given inverse couplings $\{\beta_i\}_{i=1}^R$ with estimated free energies $f_{\beta_i}$, we may reliably estimate $\langle O \rangle_\beta$ [76].

### B.2    Bootstrap error estimation

In our analysis, we estimate the error via *bootstrapping*, i.e., resampling to estimate the standard deviation of the sampling distribution of a statistic. More precisely, given a set $S$ consisting of $N$ samples, we may compute a statistic $f(S)$. By drawing $N_{bs}$ samples from $S$ with replacement, or *resampling*, one may generate $N_{bs}$ new sets $\{S_i\}_{i=1}^{N_{bs}}$, each consisting of $N$ samples, and subsequently compute the statistic $f$ on each respectively $\{f(S_i)\}_{i=1}^{N_{bs}}$. As $N_{bs} \to \infty$, the sampling distribution of the statistic $f$ on the new bootstrapped samples $\{f(S_i)\}$ approaches the sampling distribution of the statistic $f$ on the original set of samples $f(S)$. In our case, we would like to estimate the *standard error*, i.e., the standard deviation of a statistic $f$'s sampling distribution

$$\sigma_f \approx \left[ \frac{1}{N_{bs} - 1} \sum_{i=1}^{N_{bs}} (f(S_i) - \langle f(S_i) \rangle)^2 \right]^{\frac{1}{2}}. \tag{B.2}$$

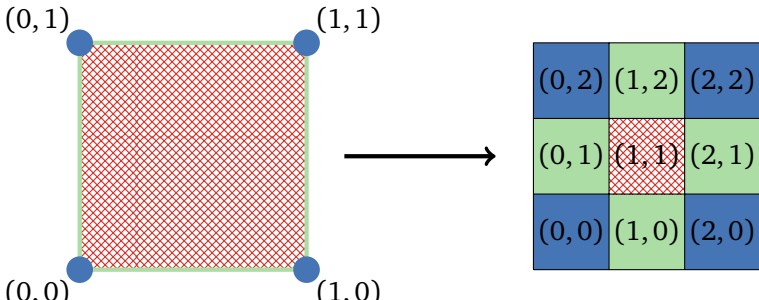

Figure 8: **(left)** On the lattice, the natural interpretation of vertices, edges and squares. **(right)** The respective $n$-cubes stored in the cubical complex data structure as a bitmap. **(explanation)** In this 2-dimensional analogue, we have a square (red hatching) bounded by four edges (green) each of which are respectively bounded by vertices (blue) at lattice sites $(0,0)$, $(0,1)$, $(1,1)$, $(1,0)$. To store this data in a 2-dimensional cubical complex data structure, we use the following coordinate system: lattice sites are mapped to vertices as $(0,0) \mapsto (0,0)$, $(0,1) \mapsto (0,2)$, $(1,1) \mapsto (2,2)$ and $(1,0) \mapsto (2,0)$; edges are are stored at $(1,0)$, $(0,1)$, $(1,2)$ and $(2,1)$; the square is stored at $(1,1)$.

## B.3 Computing Betti numbers

The dual lattice $\Lambda^*$ determines a finite partitioning of the space-time torus $T^4$ into cubes: vertices, edges, squares, 3-cubes, and 4-cubes.[4] The set of all cubes (of all dimensions) can be conveniently indexed by the set $\{0, \ldots, 2L-1\}^4$ as follows: an even number $2i$ corresponds to the points $i \in \{0 \ldots, 2L-1\}$, and an odd number $2i+1$ corresponds to the interval $[i, i+1]$. A tuple $(i_1, i_2, i_3, i_4) \in \{0, \ldots, 2L-1\}^4$ then corresponds to a cube in $T^4$ by taking the Cartesian product of the corresponding points/intervals. See Figure 8 for a 2-dimensional illustration.

This is an example of a *cubical complex with periodic boundary conditions*, which is a data structure implemented by the GUDHI library [77]. A cubical subcomplex is simply a subset of these cubes, subject to the condition that a $k$-cube being present implies that all of the cubes of its boundary are also present. Thus a subcomplex is specified by a $(2L) \times (2L) \times (2L) \times (2L)$ array containing values 0=present, 1=absent.

A 1-dimensional cubical complex is a directed graph, and so the homology of the graph can be computed via cubical complex homology. A monopole current network is a 1-dimensional subcomplex of $\Lambda^*$, and so cubical complexes provide a convenient data structure for representing these graphs. This allows us to employ accessible and optimised implementations of TDA algorithms written specifically to deal with cubical complexes. The algorithm for computing the Betti numbers of a cubical complex has computational complexity $O(n^3)$ where $n$ is the number of cubes in the complex. It involves Gaussian elimination of the boundary matrix. Often this boundary matrix is sparse and thus the computation is typically closer to $O(n^2)$. For more details on the theory of cubical homology, useful references are [78, 79].

In this study, we computed cubical homology using the giotto-tda python library [73], which is a python wrapper for the GUDHI C++ library. For details, see the accompanying software release in Ref. [72]. Note that other optimised libraries exist for computing TDA, e.g., Ripser python and C++ packages in Refs. [80, 81].

---

[4]More precisely, one may refer to the lattice $\Lambda^*$ as a *stratification* of $T^4$ where the *strata* are the vertices, edges, squares, 3-cubes, and 4-cubes.

## C Finite-size scaling analysis: Further details

In this Appendix, we provide an analysis that checks the robustness of the fit from Equation (12) by varying $k_{\max}$ and the range of values. We compute the reduced chi-squared statistic $\chi_{\nu}^2$ as a measure of goodness of fit. We present our results for $E$, $\rho_0$ and $\rho_1$ in Table 4, Table 5 and Table 6 respectively. Our chosen fits are in bold and corresponds to the parameter choice for which $\chi_{\nu}^2$ takes the closest value to one.

Table 4: Finite-size scaling analysis for $E$.

| $L$ | $k_{\max}$ | $\chi_{\nu}^2$ | $\beta_c$ |
|---|---|---|---|
| $10, 11, 12$ | **1** | **0.760** | **1.01071(3)** |
| $8, 9, 10, 11, 12$ | 1 | 6.563 | 1.01061(2) |
| | 2 | 0.185 | 1.01078(5) |
| | 3 | 0.412 | 1.0107(1) |
| $6, 7, 8, 9, 10, 11, 12$ | 1 | 154.216 | 1.01021(2) |
| | 2 | 4.222 | 1.01066(3) |
| | 3 | 0.140 | 1.01078(4) |
| | 4 | 0.208 | 1.01080(8) |
| | 5 | 0.417 | 1.01080(8) |

Table 5: Finite-size scaling analysis for $\rho_0$.

| $L$ | $k_{\max}$ | $\chi_{\nu}^2$ | $\beta_c$ |
|---|---|---|---|
| $10, 11, 12$ | 1 | 2.685 | 1.01070(3) |
| $8, 9, 10, 11, 12$ | 1 | 6.152 | 1.01061(4) |
| | **2** | **0.821** | **1.01076(6)** |
| | 3 | 1.724 | 1.0108(1) |
| $6, 7, 8, 9, 10, 11, 12$ | 1 | 135.044 | 1.01022(3) |
| | 2 | 3.368 | 1.01066(7) |
| | 3 | 0.545 | 1.01076(9) |
| | 4 | 0.782 | 1.0108(1) |
| | 5 | 1.563 | 1.0108(1) |

Table 6: Finite-size scaling analysis for $\rho_1$.

| $L$ | $k_{\max}$ | $\chi_\nu^2$ | $\beta_c$ |
|---|---|---|---|
| $10, 11, 12$ | 1 | 0.082 | 1.01079(4) |
| $8, 9, 10, 11, 12$ | 1 | 1.667 | 1.01072(3) |
| | **2** | **1.016** | **1.01076(6)** |
| | 3 | 0.183 | 1.0109(1) |
| $6, 7, 8, 9, 10, 11, 12$ | 1 | 40.103 | 1.0105(1) |
| | 2 | 2.405 | 1.01072(9) |
| | 3 | 0.503 | 1.01083(7) |
| | 4 | 1.138 | 1.0107(1) |
| | 5 | 2.276 | 1.0107(1) |

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
