# Peer review of "Topological Data Analysis of Monopole Current Networks in $U(1)$ Lattice Gauge Theory"

_SciPost Physics, doi:SciPost Phys. 17, 100 (2024)_

## Round 2 · Referee Report · Anonymous (Referee 1) · 2024-5-17

Strengths

1. Interesting application of topological data analysis (TDA) to lattice gauge theory.
2. Clear quantitative predictions using homology.
3. State-of-the-art lattice methods.

Weaknesses

1. TDA part not on the same quality level as the lattice part.

Report

The present work provides an interesting application of TDA, a relatively new field of data-driven applied topology, to the deconfinement phase transition of U(1) lattice gauge theory. Building on older results of [Kerler et al., Phys. Lett. B 348, 1995], the authors employ Betti numbers to characterize monopole current networks. This fresh approach allows them to obtain estimates for the pseudo-critical inverse coupling at infinite volume, which agree with those computed from average plaquette actions and have error bars of similar size.

The actual lattice gauge theory part of the paper is profound and well written. In particular, this applies to the introduction, to the non-TDA descriptions of the phase transition in the studied U(1) lattice gauge theory and the related lattice computations (i.e., Secs. 1 and 2).

The authors publish the utilized software and the investigated data using the platform Zenodo. After a superficial inspection, the code and data provision for the work seems exemplary (I did not re-run the simulations and scripts).

This brings me to my major points of critique:

1. Towards the end of the abstract and in the introduction, the authors claim that their approach can be generalized to investigate monopoles in non-Abelian gauge theories. Yet, the authors do not provide any further discussion of this. At least in the conclusions, the authors should provide a paragraph on the envisioned generalization of their analysis pipeline to the non-Abelian case.
2. At the beginning of Sec. 2.2, the authors introduce the nomenclature that the low-$\beta$ phase is called hot and the large-$\beta$ phase is called cold, based on the analogy of $\beta$ with the role of the inverse temperature. Even if this argument has been employed in older papers [e.g., Kerler et al., Phys. Lett. B 348, 1995], I find it misleading: if physical temperatures for lattice gauge theories are regarded, the inverse nomenclature is more appropriate. Indeed, even U(1) lattice gauge theory is confining in the low physical temperature regime (low $\beta$) and deconfines at a larger physical temperature [see, e.g., Svetitsky & Yaffe, Nucl. Phys. B 210, 1982]. I recommend that throughout their paper, the authors change the nomenclature to a solely $\beta$-based one, entirely refraining from calling the regimes hot and cold, but potentially including a sentence on the correlation between $\beta$ and the physical temperature.
3. At multiple points in their manuscript, the authors mention how beneficial it is to consider the investigated monopole current networks as directed graphs. Yet barely any details are provided in this regard: in the beginning of Sec. 3, the authors provide a short paragraph on the definition of the monopole current graph $X_j$, which is complemented by another brief paragraph on the definition of the Betti numbers of $X_j$. In the acknowledgements, they mention that the software package giotto-tda is utilized for the computation of the Betti numbers, and Appendix C provides some background on homology.
To me, it would appear natural to introduce a separate subsection 3.1, which is devoted to the introduction of the monopole current graph and to the introduction of its Betti numbers, potentially accompanied by intuitive illustrations to improve the readability for the readers not acquainted with TDA. This subsection should also include more details on how specifically giotto-tda is used to compute the Betti numbers, regarding the underlying monopole current graph as a directed graph. The latter is not clear to me who is accustomed with TDA, and likely even much less clear to the general readership of SciPost Physics. Including such a subsection and spending more time to carefully introduce the employed TDA observables and computational pipeline could substantially improve the readability of the TDA-related part of the manuscript as well as emphasize the promise of TDA for lattice simulations.
4. The description of the TDA-based observables shown in Figs. 3-5 with the inferred critical inverse couplings given in Tables 2 and 3 is described on only a bit more than half a page in Sec. 3. I recommend making it a longer subsection 3.2, in which further details are provided, and if possible, also more intuitive explanations for the behavior of the Betti number densities. For instance, the authors write that the maximum of $\rho_{b_0}$ is realized when the large percolating network breaks up into smaller networks. Can the authors provide arguments underlying this claim, at least mentioning that this can be a posteriori justified by the agreement of the pseudo-critical inverse couplings? Furthermore, the authors write that $\rho_{b_0}$ declines for larger $\beta$-values, since the energy cost for generating a monopole current loop is expensive. Can this be described less vague? Fig. 4 is described only in two sentences, which certainly deserves a longer discussion including a probably intuitively accessible interpretation based on the behavior of the (percolating) monopole current networks (referring also to Sec. 2).

Smaller remarks:

1. In Sec. 2.2, it is written that monopole current conservation and the periodic, untwisted boundary conditions imply the total charge of the current loops is zero. I assume there is a topological argument for this, based on homotopy classes of maps from the lattice 4-torus to the set $\{0,\pm 1, \pm 2\}$ or a variant of the latter (maybe I’m missing a piece of knowledge here). For better accessibility to their work also for non-lattice experts, can the authors provide the argument for this (potentially in a footnote)?
2. I’m wondering about the prefactors in Eq. (8): while the $1/(4\pi)$ prefactor in the first equality is consistent with [Kerler et al., Phys. Lett. B 348, 1995], the next line should contain a prefactor $-1/2$. Then Eq. (8) would be also consistent with Eqs. (13) and (14).
3. Even if the extrapolated infinite-volume, pseudo-critical inverse couplings computed for zeroth and first Betti numbers agree, I’m curious to learn whether the authors have an interpretation for the pseudo-critical inverse coupling computed from $\rho_{b_1}$ being consistently smaller than the one computed from $\rho_{b_0}$ for finite lattices. Given that this is a very small effect, I would not expect this to be discussed in the manuscript.
4. In Figs. 3-5 ensemble averages for the densities $\rho_{b_k}$ are shown. In the main text, no ensemble averages are indicated. Can the notation be made consistent in this regard?
5. In Sec. 3 and Appendix C.2 it is stated that the Betti numbers are independent from the coefficient field. Targeting the non-mathematician readership of SciPost Physics, I believe that a corresponding reference for this fact or a rough explanation in a footnote would be beneficial.
6. Throughout Figs. 3-5, subfigures (b) show zoom-ins of subfigures (a). I am confident that the authors can highlight this fact visually better, so that it is more intuitively clear, e.g., by means of diagonal lines and a square in subfigures (a), indicating the displayed zoom-in.
7. The authors mention in Sec. 3 twice that the reweighted variance curves for $\rho_{b_k}$, $k=0,1$, peak at the critical point (partially obscured). Can the authors maybe provide these figures as distinct plots, potentially shrinking the prominent zoom-ins of subfigures (b) and showing the reweighted variance curves in addition?

To conclude, the main results of the work concerning the application of TDA to monopole current networks in U(1) lattice gauge theory are indeed valid and new, and moreover provide another fruitful study that showcases the potential of TDA for physics applications. Yet, I feel that a few points deserve more attention and in particular Sec. 3 on the actual topological data analysis requires a major rewriting and expansion. For this reason, even if the paper and its results fit into SciPost Physics, in its current form I recommend a major revision of the manuscript.

Requested changes

The requested changes have been mentioned in the report and concern mostly but not exclusively Sec. 3.

Recommendation

Ask for major revision

  • validity: good
  • significance: good
  • originality: high
  • clarity: high
  • formatting: good
  • grammar: excellent

Author:  Xavier Crean  on 2024-07-19  [id 4638]

(in reply to Report 1 on 2024-05-17)

In this report, we respond to comments made by the first referee (henceforth ‘Ref1’). Where similar comments were made by the second referee (henceforth ‘Ref2’), the response has been combined. For the referee’s/editor’s convenience, we have attached a pdf showing explicitly the differences between the two versions.

Referee’s Comments: Ref2: 1. It is mentioned in the abstract (and briefly in the introduction) that the present approach can be generalised to study Abelian monopoles in non-Abelian LGT. However, there is no comment throughout the rest of the manuscript on how this could be done and which major technical difficulties one would need to overcome in this case (if any). I think it would be nice if the authors could elaborate on this point, for example, in Sec. 4, as it seems to be a very interesting outlook. Ref1: 1. Towards the end of the abstract and in the introduction, the authors claim that their approach can be generalized to investigate monopoles in non-Abelian gauge theories. Yet, the authors do not provide any further discussion of this. At least in the conclusions, the authors should provide a paragraph on the envisioned generalization of their analysis pipeline to the non-Abelian case. Author’s Response: We have included a paragraph in the conclusion discussing future applications of our methodology to the non-Abelian case.

Referee’s Comments: Ref1: 2. At the beginning of Sec. 2.2, the authors introduce the nomenclature that the low-β phase is called hot and the large-β phase is called cold, based on the analogy of β with the role of the inverse temperature. Even if this argument has been employed in older papers [e.g., Kerler et al., Phys. Lett. B 348, 1995], I find it misleading: if physical temperatures for lattice gauge theories are regarded, the inverse nomenclature is more appropriate. Indeed, even U(1) lattice gauge theory is confining in the low physical temperature regime (low β) and deconfines at a larger physical temperature [see, e.g., Svetitsky & Yaffe, Nucl. Phys. B 210, 1982]. I recommend that throughout their paper, the authors change the nomenclature to a solely β-based one, entirely refraining from calling the regimes hot and cold, but potentially including a sentence on the correlation between β and the physical temperature. Author’s Response: Throughout the paper, we have changed the nomenclature from hot/cold phase to low-β/high-β phase.

Referee’s Comments: Ref2: 3. I appreciate that the authors try not to obscure the discussion in the main text with technical aspects. However, since Betti numbers play a central role in their analysis, I think that the authors should consider to include Appendix C as a part of Sec. 3 in the main text. Ref1: 3. At multiple points in their manuscript, the authors mention how beneficial it is to consider the investigated monopole current networks as directed graphs. Yet barely any details are provided in this regard: in the beginning of Sec. 3, the authors provide a short paragraph on the definition of the monopole current graph Xj, which is complemented by another brief paragraph on the definition of the Betti numbers of Xj. In the acknowledgements, they mention that the software package giotto-tda is utilized for the computation of the Betti numbers, and Appendix C provides some background on homology. To me, it would appear natural to introduce a separate subsection 3.1, which is devoted to the introduction of the monopole current graph and to the introduction of its Betti numbers, potentially accompanied by intuitive illustrations to improve the readability for the readers not acquainted with TDA. This subsection should also include more details on how specifically giotto-tda is used to compute the Betti numbers, regarding the underlying monopole current graph as a directed graph. The latter is not clear to me who is accustomed with TDA, and likely even much less clear to the general readership of SciPost Physics. Including such a subsection and spending more time to carefully introduce the employed TDA observables and computational pipeline could substantially improve the readability of the TDA-related part of the manuscript as well as emphasize the promise of TDA for lattice simulations. Author’s Response: There are several changes we have made here: 1. We have moved the Appendix C that describes background on homology and Betti numbers to a new Subsection 3.1 and created another new subsection 3.2 called ‘Analysis of Monopole Current Networks’ that contains content from the original submission. 2. We have created a new Appendix B.3 (a Subsection in the Appendix 3 on Methods) that includes additional details on our computational pipeline. We have gone into much more detail on how we actually compute Betti numbers of the directed graph using giotto-tda. This entails mapping graphs to a 1-dimensional cubical complex with periodic boundary conditions (a data structure that is implemented by the GUDHI library); we have included an explanatory figure. 3. In Subsubsection 3.2.1, we have rephrased the description of how we map monopole current networks to directed graphs. The main point we are trying to get across here is that for the definition of graph homology to be consistent there must be an orientation associated with each edge in the graph; however, this orientation is arbitrary and the resulting homology is independent of the choice. It does not have to be aligned with the orientation of the currents. We have introduced graph homology as vehicle to explain how we compute the homological invariants associated to current networks; we may well have introduced this in the setting of cubical complexes (which is used in the computational pipeline); we make the choice of the graph homology to avoid obscuring the explanation with mathematical technicalities (involved in defining cubical complexes with periodic boundary conditions) that are not necessarily key to understanding what we have computed. As mentioned above, we have included some details on the computational pipeline including cubical complexes in the new Appendix B.3.

Referee’s Comments: Ref1: 4. The description of the TDA-based observables shown in Figs. 3-5 with the inferred critical inverse couplings given in Tables 2 and 3 is described on only a bit more than half a page in Sec. 3. I recommend making it a longer subsection 3.2, in which further details are provided, and if possible, also more intuitive explanations for the behavior of the Betti number densities. For instance, the authors write that the maximum of ρb0 is realized when the large percolating network breaks up into smaller networks. Can the authors provide arguments underlying this claim, at least mentioning that this can be a posteriori justified by the agreement of the pseudo-critical inverse couplings? Furthermore, the authors write that ρb0 declines for larger β-values, since the energy cost for generating a monopole current loop is expensive. Can this be described less vague? Fig. 4 is described only in two sentences, which certainly deserves a longer discussion including a probably intuitively accessible interpretation based on the behavior of the (percolating) monopole current networks (referring also to Sec. 2). Author’s Response: The picture we describe of a large percolating network in the low beta phase and small distinct networks in the high beta phase is based on the accepted physical picture from the literature. We have added some relevant references to aid the reader. Our comment about the large network breaking into smaller networks as \rho_0 obtains its maximum is a statement that in the critical region the behaviour of \rho_0 corresponds with the accepted physical picture from the literature. We have rephrased the language to clarify the point that at large \beta values sampled configurations are less likely to contain monopole current loops.

Referee’s Comments: Ref2: 4. In Sec. 3, when introducing Betti numbers, it is mentioned that "We use the coefficient field Z2 but, since this is a 1-dimensional complex, the resulting Betti numbers are independent of the coefficient field." I think that adding a relevant reference here is in order. Ref1: 5. In Sec. 3 and Appendix C.2 it is stated that the Betti numbers are independent from the coefficient field. Targeting the non-mathematician readership of SciPost Physics, I believe that a corresponding reference for this fact or a rough explanation in a footnote would be beneficial. Author’s Response: We have included an explanation/reference in a footnote. Any graph is homotopy equivalent to a bouquet of circles and the Universal Coefficient Theorem says that the rank of the homology groups of a bouquet of circles are independent of the coefficient group A where A is an Abelian group.

Referee’s Comments: Ref2: 5. In Sec. 3, the authors talk about the reweighted variance curves for the observables ρb0 and ρb1. Including examples of such curves (as figures, for instance, in Appendix B.1) could help to understand better the explanation provided there. Ref1: 7. The authors mention in Sec. 3 twice that the reweighted variance curves for ρbk, k=0,1 peak at the critical point (partially obscured). Can the authors maybe provide these figures as distinct plots, potentially shrinking the prominent zoom-ins of subfigures (b) and showing the reweighted variance curves in addition? Author’s Response: We have included the variance plots in the main body of the text with appropriate inset zoom-ins to highlight the peaks in the variance (including error bars) in the critical region.

Referee’s Comments: Ref1: In Sec. 2.2, it is written that monopole current conservation and the periodic, untwisted boundary conditions imply the total charge of the current loops is zero. I assume there is a topological argument for this, based on homotopy classes of maps from the lattice 4-torus to the set {0,±1,±2} or a variant of the latter (maybe I’m missing a piece of knowledge here). For better accessibility to their work also for non-lattice experts, can the authors provide the argument for this (potentially in a footnote)? Author’s Response: We have included a sentence explaining this. Due to the lattice equivalent of Gauss’ law for magnetism, the total monopole charge in the configuration must sum to zero. Loops that wrap around the hyper-torus in a given direction contribute a non-zero charge that can only be cancelled by an oppositely oriented wrapping loop (in the same direction).

Referee’s Comments: Ref1: 2. I’m wondering about the prefactors in Eq. (8): while the 1/(4π) prefactor in the first equality is consistent with [Kerler et al., Phys. Lett. B 348, 1995], the next line should contain a prefactor −1/2. Then Eq. (8) would be also consistent with Eqs. (13) and (14). Author’s Response: This was a typo that has now been corrected: Equation (8) now includes the prefactor -1/2.

Referee’s Comments: Ref1: 3. Even if the extrapolated infinite-volume, pseudo-critical inverse couplings computed for zeroth and first Betti numbers agree, I’m curious to learn whether the authors have an interpretation for the pseudo-critical inverse coupling computed from ρb1 being consistently smaller than the one computed from ρb0 for finite lattices. Given that this is a very small effect, I would not expect this to be discussed in the manuscript. Author’s Response: We have added a footnote highlighting that whilst E, \rho_0 and \rho_1 are not equal for finite size L, they converge in the physically significant infinite volume limit.

Referee’s Comments: Ref1: 4. In Figs. 3-5 ensemble averages for the densities ρbk are shown. In the main text, no ensemble averages are indicated. Can the notation be made consistent in this regard? Author’s Response: Throughout the paper, the notation has been clarified and updated.

Referee’s Comments: Ref1: 6. Throughout Figs. 3-5, subfigures (b) show zoom-ins of subfigures (a). I am confident that the authors can highlight this fact visually better, so that it is more intuitively clear, e.g., by means of diagonal lines and a square in subfigures (a), indicating the displayed zoom-in. Author’s Response: Inset zoom-ins have been added to the plots to highlight the behaviour in the critical region more clearly.

Attachment:

U1_TDA_arXiv_v3_revision_tnqCm9a.pdf

---

## Round 2 · Referee Report · Anonymous (Referee 2) · 2024-5-31

Strengths

1. Very well written.
2. Clear and sound presentation of the problem and the results.
3. High level of interpretability of the observables defined through topological data analysis.

Weaknesses

In my opinion, this work does not present any major weaknesses, and I consider that all the suggested changes can be addressed at this stage.

Report

This work explores the use of topological data analysis (TDA) to characterise topological aspects of a 4-dimensional pure compact U(1) lattice gauge theory (LGT), relating them to the phase structure of the model. The main contribution of this work is to show that relatively simple and highly interpretable topological observables, built from the homology groups of monopole current networks, allow for a quantitative estimation of the critical inverse coupling of the theory via finite-size scaling analysis. This is possible as monopole current networks can be regarded as directed graphs and, therefore, described using notions of algebraic topology within the framework of TDA. While topological characterisations of this problem have been proposed previously in the literature, the present work illustrates in a novel way the usefulness of TDA for problems relevant to LGT and beyond, with the potential for follow-up studies. As this type of data analysis is attracting growing attention in the physics community, I believe that this work will be of interest to a broad readership of the SciPost Physics.

The paper generally satisfies the acceptance criteria of SciPost Physics. However, before I can recommend for its publication, I would ask the authors to consider the changes mentioned below.

Requested changes

1. It is mentioned in the abstract (and briefly in the introduction) that the present approach can be generalised to study Abelian monopoles in non-Abelian LGT. However, there is no comment throughout the rest of the manuscript on how this could be done and which major technical difficulties one would need to overcome in this case (if any). I think it would be nice if the authors could elaborate on this point, for example, in Sec. 4, as it seems to be a very interesting outlook.

2. The authors correctly acknowledge and summarise previous works exploring topological aspects of monopoles currents and Dirac sheets in Abelian LGT. In particular, Ref. 10 by Kerler et al., which is mostly based on the topological analysis of Dirac sheets, is taken as a reference point for the present work. However, a previous work also by Kerler et al., namely Ref. 22, provides a topological characterisation of networks of current lines, which is based on their fundamental homotopy group. This can be regarded as a further application of TDA to characterise monopole current networks of the system under study. Therefore, I think it would be relevant to mention more explicitly the work in Ref. 22. This could emphasise possible advantages of the present approach (e.g., it is my understanding that properties based on homology are, in general, easier to compute than those based on homotopy, etc.).

3. I appreciate that the authors try not to obscure the discussion in the main text with technical aspects. However, since Betti numbers play a central role in their analysis, I think that the authors should consider to include Appendix C as a part of Sec. 3 in the main text.

4. In Sec. 3, when introducing Betti numbers, it is mentioned that "We use the coefficient field $\mathbb{Z}_2$ but, since this is a 1-dimensional complex, the resulting Betti numbers are independent of the coefficient field." I think that adding a relevant reference here is in order.

5. In Sec. 3, the authors talk about the reweighted variance curves for the observables $\rho_{b_0}$ and $\rho_{b_1}$. Including examples of such curves (as figures, for instance, in Appendix B.1) could help to understand better the explanation provided there.

6. I would suggest to include in the title the wording "Monopole Current Networks" instead of just "Monopoles", since this would be a more accurate description.

7. In Appendix B.2, I believe that the samples used in the bootstrap analysis are of size $N$. This could be mentioned explicitly.

Recommendation

Ask for minor revision

  • validity: high
  • significance: high
  • originality: high
  • clarity: top
  • formatting: excellent
  • grammar: perfect

Author:  Xavier Crean  on 2024-07-19  [id 4637]

(in reply to Report 2 on 2024-05-31)

In this report, we respond to comments made by the second referee (henceforth ‘Ref2’). Where similar comments were made by the first referee (henceforth ‘Ref1’), the response has been combined. For the referee’s/editor’s convenience, we have attached a pdf showing explicitly the differences between the two versions.

Referee’s Comments: Ref2: 1. It is mentioned in the abstract (and briefly in the introduction) that the present approach can be generalised to study Abelian monopoles in non-Abelian LGT. However, there is no comment throughout the rest of the manuscript on how this could be done and which major technical difficulties one would need to overcome in this case (if any). I think it would be nice if the authors could elaborate on this point, for example, in Sec. 4, as it seems to be a very interesting outlook. Ref1: 1. Towards the end of the abstract and in the introduction, the authors claim that their approach can be generalized to investigate monopoles in non-Abelian gauge theories. Yet, the authors do not provide any further discussion of this. At least in the conclusions, the authors should provide a paragraph on the envisioned generalization of their analysis pipeline to the non-Abelian case. Author’s Response: We have included a paragraph in the conclusion discussing future applications of our methodology to the non-Abelian case.

Referee’s Comments: Ref2: 3. I appreciate that the authors try not to obscure the discussion in the main text with technical aspects. However, since Betti numbers play a central role in their analysis, I think that the authors should consider to include Appendix C as a part of Sec. 3 in the main text. Ref1: 3. At multiple points in their manuscript, the authors mention how beneficial it is to consider the investigated monopole current networks as directed graphs. Yet barely any details are provided in this regard: in the beginning of Sec. 3, the authors provide a short paragraph on the definition of the monopole current graph Xj, which is complemented by another brief paragraph on the definition of the Betti numbers of Xj. In the acknowledgements, they mention that the software package giotto-tda is utilized for the computation of the Betti numbers, and Appendix C provides some background on homology. To me, it would appear natural to introduce a separate subsection 3.1, which is devoted to the introduction of the monopole current graph and to the introduction of its Betti numbers, potentially accompanied by intuitive illustrations to improve the readability for the readers not acquainted with TDA. This subsection should also include more details on how specifically giotto-tda is used to compute the Betti numbers, regarding the underlying monopole current graph as a directed graph. The latter is not clear to me who is accustomed with TDA, and likely even much less clear to the general readership of SciPost Physics. Including such a subsection and spending more time to carefully introduce the employed TDA observables and computational pipeline could substantially improve the readability of the TDA-related part of the manuscript as well as emphasize the promise of TDA for lattice simulations. Author’s Response: There are several changes we have made here: 1. We have moved the Appendix C that describes background on homology and Betti numbers to a new Subsection 3.1 and created another new subsection 3.2 called 'Analysis of Monopole Current Networks' that contains content from the original submission. 2. We have created a new Appendix B.3 (a Subsection in the Appendix 3 on Methods) that includes additional details on our computational pipeline. We have gone into much more detail on how we actually compute Betti numbers of the directed graph using giotto-tda. This entails mapping graphs to a 1-dimensional cubical complex with periodic boundary conditions (a data structure that is implemented by the GUDHI library); we have included an explanatory figure. 3. In Subsubsection 3.2.1, we have rephrased the description of how we map monopole current networks to directed graphs. The main point we are trying to get across here is that for the definition of graph homology to be consistent there must be an orientation associated with each edge in the graph; however, this orientation is arbitrary and the resulting homology is independent of the choice. It does not have to be aligned with the orientation of the currents. We have introduced graph homology as vehicle to explain how we compute the homological invariants associated to current networks; we may well have introduced this in the setting of cubical complexes (which is used in the computational pipeline); we make the choice of the graph homology to avoid obscuring the explanation with mathematical technicalities (involved in defining cubical complexes with periodic boundary conditions) that are not necessarily key to understanding what we have computed. As mentioned above, we have included some details on the computational pipeline including cubical complexes in the new Appendix B.3.

Referee’s Comments: Ref2: 2. The authors correctly acknowledge and summarise previous works exploring topological aspects of monopoles currents and Dirac sheets in Abelian LGT. In particular, Ref. 10 by Kerler et al., which is mostly based on the topological analysis of Dirac sheets, is taken as a reference point for the present work. However, a previous work also by Kerler et al., namely Ref. 22, provides a topological characterisation of networks of current lines, which is based on their fundamental homotopy group. This can be regarded as a further application of TDA to characterise monopole current networks of the system under study. Therefore, I think it would be relevant to mention more explicitly the work in Ref. 22. This could emphasise possible advantages of the present approach (e.g., it is my understanding that properties based on homology are, in general, easier to compute than those based on homotopy, etc.). Author’s Response: We have more explicitly mentioned Kerler et al. Ref. [22] in the introduction. We have also included a comment on the fact that the phase transition was found, e.g., in Ref. [29], to exist on a lattice representation of the 4-sphere where non-trivial winding is not possible since all loops are contractible.

Referee’s Comments: Ref2: 4. In Sec. 3, when introducing Betti numbers, it is mentioned that "We use the coefficient field Z2 but, since this is a 1-dimensional complex, the resulting Betti numbers are independent of the coefficient field." I think that adding a relevant reference here is in order. Ref1: 5. In Sec. 3 and Appendix C.2 it is stated that the Betti numbers are independent from the coefficient field. Targeting the non-mathematician readership of SciPost Physics, I believe that a corresponding reference for this fact or a rough explanation in a footnote would be beneficial. Author’s Response: We have included an explanation/reference in a footnote. Any graph is homotopy equivalent to a bouquet of circles and the Universal Coefficient Theorem says that the rank of the homology groups of a bouquet of circles are independent of the coefficient group A where A is an Abelian group.

Referee’s Comments: Ref2: 5. In Sec. 3, the authors talk about the reweighted variance curves for the observables ρb0 and ρb1. Including examples of such curves (as figures, for instance, in Appendix B.1) could help to understand better the explanation provided there. Ref1: 7. The authors mention in Sec. 3 twice that the reweighted variance curves for ρbk, k=0,1 peak at the critical point (partially obscured). Can the authors maybe provide these figures as distinct plots, potentially shrinking the prominent zoom-ins of subfigures (b) and showing the reweighted variance curves in addition? Author’s Response: We have included the variance plots in the main body of the text with appropriate inset zoom-ins to highlight the peaks in the variance (including error bars) in the critical region.

Referee’s Comments: Ref2: 6. I would suggest to include in the title the wording "Monopole Current Networks" instead of just "Monopoles", since this would be a more accurate description. Author’s Response: The title has been changed to “Topological Data Analysis of Monopole Current Networks in U(1) Lattice Gauge Theory”.

Referee’s Comments: Ref2: 7. In Appendix B.2, I believe that the samples used in the bootstrap analysis are of size N. This could be mentioned explicitly. Author’s Response: We have a made a small edit to the text in Appendix B.2 to highlight the fact that each set S_i, generated by the resampling, consists of N samples.

Attachment:

U1_TDA_arXiv_v3_revision.pdf

---

## Round 3 · Referee Report · Anonymous (Referee 1) · 2024-8-1

Strengths

  1. Interesting application of topological data analysis (TDA) to lattice gauge theory.
  2. Clear quantitative predictions using homology.
  3. State-of-the-art lattice methods.

Report

The authors have provided a major revision of the manuscript, which, from my point of view, forms a substantial improvement compared to its first version. They suitably addressed all my points of critique. In particular, the presentation of the TDA methodology and the discussion of the related results has been significantly improved. The inclusion of the variance curves provides a good addition.

Solely, the manuscript appears now somewhat technical at some points to me, in particular regarding the concepts from algebraic topology. A sentence on the relation between chain complexes and graphs, based on lattice gauge theory configurations, might help guiding the physical intuition in Sec. 3.1.2. Furthermore, in Sec. 3.1.2 I appreciate the argument regarding coefficient field independence of the Betti numbers, but maybe the sentence including the universal coefficient theorem can be included in footnote 2 as well? I am doubtful, whether the anticipated readership of SciPost Physics is aware of the universal coefficient theorem; and the statement of coefficient field independence would then still be included in the main text. Finally, Appendix B.3 including the related new Fig. 8 adds value to the manuscript, but I am confident that the concept of a stratification can be omitted, which is certainly unclear to many physicists.

Yet, I recommend publication of the manuscript in SciPost Physics.

Requested changes

The minor change recommendations, which are to be viewed as suggestions only, have been highlighted in the report.

Recommendation

Publish (meets expectations and criteria for this Journal)

---

## Round 3 · Referee Report · Anonymous (Referee 2) · 2024-8-5

Strengths

  1. Well written.
  2. Clear and sound presentation of the problem and the results.
  3. High level of interpretability of the observables defined through topological data analysis.

Weaknesses

I do not see any weaknesses in the current form of the manuscript.

Report

The revised manuscript reads well and addresses the points that I raised during the first round of review. Therefore, I recommend its publication in SciPost Physics.

Below are a couple of details for the authors to consider:

  • In the second footnote, I believe that where the authors write $k>2$, it should read $k \ge 2$.
  • The label on the y-axis in Fig. 5 should be $\rho_1$ instead of $\rho_0$.
  • While the peak signal in Figs. 4 and 6 can be clearly seen, the scattered symbols make these plots a bit difficult to read in detail. Adding lines as a guides to the eye or increasing the size of the markers could be helpful.

Recommendation

Publish (easily meets expectations and criteria for this Journal; among top 50%)

---

## Round 3 · Author Response

Dear Editor,

Please accept the resubmission of our paper 'Topological Data Analysis of Monopole Current Networks in U(1) Lattice Gauge Theory'.

Yours sincerely,
The Authors

---

## Round 3 · List of Changes

Changes are in the format (1) brief summary of the referee request, (2) change that the authors have made.

  1. More about future non-Abelian application: We have included a paragraph in the conclusion discussing future applications of our methodology to the non-Abelian case.

  2. Change hot/cold nomenclature to low-beta/high-beta: Throughout the paper, we have changed the nomenclature from hot/cold phase to low-β/high-β phase.

  3. Move subsection introducing the Betti numbers to subsection 3.1 and expand section 3: There are several changes we have made here: 3.1 We have moved the Appendix C that describes background on homology and Betti numbers to a new Subsection 3.1 and created another new subsection 3.2 called Analysis of Monopole Current Networks that contains content from the original submission. 3.2 We have created a new Appendix B.3 (a Subsection in the Appendix 3 on Methods) that includes additional details on our computational pipeline. We have gone into much more detail on how we actually compute Betti numbers of the directed graph using giotto-tda. This entails mapping graphs to a 1-dimensional cubical complex with periodic boundary conditions (a data structure that is implemented by the GUDHI library); we have included an explanatory figure. 3.3 In Subsubsection 3.2.1, we have rephrased the description of how we map monopole current networks to directed graphs.

  4. Clarify explanation of large network breaking into smaller networks The picture we describe of a large percolating network in the low beta phase and small distinct networks in the high beta phase is based on the accepted physical picture from the literature. We have added some relevant references to aid the reader. Our comment about the large network breaking into smaller networks as \rho_0 obtains its maximum is a statement that in the critical region the behaviour of \rho_0 corresponds with the accepted physical picture from the literature. We have rephrased the language to clarify the point that at large \beta values sampled configurations are less likely to contain monopole current loops.

  5. Reference Kerler et al more explicitly We have more explicitly mentioned Kerler et al. Ref. [22] in the introduction. We have also included a comment on the fact that the phase transition was found, e.g., in Ref. [29], to exist on a lattice representation of the 4-sphere where non-trivial winding is not possible.

  6. Add citation for graph homology independent of coefficient field We have included an explanation/reference in a footnote.

  7. Add variance plots of the observables We have included the variance plots in the main body of the text with appropriate inset zoom-ins to highlight the peaks in the variance (including error bars) in the critical region.

  8. Explain why periodically closed loops come in oppositely oriented pairs We have included a sentence explaining this.

  9. Prefactors in monopole number equation (8) This was a typo that has now been corrected: Equation (8) now includes the prefactor -1/2.

  10. Explain why betti_1 has smaller critical beta than betti_0 We have added a footnote highlighting that whilst E, \rho_0 and \rho_1 are not equal for finite size L, they converge in the physically significant infinite volume limit.

  11. Make notation consistent for density of betti_k Throughout the paper, the notation has been clarified and updated.

  12. Make zoom-in plots embedded with diagonal lines Inset zoom-ins have been added to the plots to highlight the behaviour in the critical region more clearly.

  13. Change title to highlight Monopole Current Networks The title has been changed to “Topological Data Analysis of Monopole Current Networks in U(1) Lattice Gauge Theory”.

  14. Sample number N in bootstrap error estimation We have a made a small edit to the text in Appendix B.2 to highlight the fact that each set S_i, generated by the resampling, consists of N samples.

---

## Round 4 · Author Response

Dear Editor,

Please accept the resubmission of our paper 'Topological Data Analysis of Monopole Current Networks in U(1) Lattice Gauge Theory'.

Yours sincerely,
The Authors

---

## Round 4 · List of Changes

Changes are in the format (1) brief summary of the referee request, (2) change that the authors have made.

  1. Include sentence on the relation between chain complexes and graphs: We have added a clarifying sentence to the start of Section 3.1.2.

  2. Move Universal Coefficient Theorem to footnote: The sentence about the Universal Coefficient Theorem has been moved to the footnote on page 9.

  3. Replacement of term ‘stratification’ in Appendix B.3: In Appendix B.3, in the main body of text, the term ‘stratification’ has been replaced by ‘finite partitioning’. A footnote has been added that mentions stratification for those readers who are familiar with the notion of a stratified space.

  4. Correct typos: Two typos were identified by a referee; these have been corrected.

  5. Figures 4 and 6 use larger scatter symbols and curves to guide eye: There are two changes we have made here. Firstly, for Figures 4 and 6, in the inset plots, we have increased the size of the markers and included the histogram reweighting curves used in the analysis. We believe this provides an appropriate visual guide. Note that reweighting windows have been suitably chosen to cover the peak of \chi_k points for each respective lattice size. Secondly, we have updated the captions for Figures 3, 4, 5 and 6 so that they all follow the same format of main plot description then inset plot description. The description of the plots has been updated to reflect the changes.

---

## Editorial Decision

published